# Learning Cognitive Features from Gaze Data for Sentiment and Sarcasm Classification using Convolutional Neural Network

## Abstract

Cognitive NLP systems- *i.e.*, NLP systems that make use of behavioral data - augment traditional text based features with cognitive features extracted from eye-movement patterns, EEG signals, brain-imaging *etc.*. Such extraction of features is typically manual. We contend that manual extraction of features is not good enough to tackle text subtleties that characteristically prevail in complex classification tasks like *sentiment analysis* and *sarcasm detection*, and that even the extraction and choice of features should be delegated to the learning system. We introduce a framework to automatically extract cognitive features from the *eye-movement / gaze* data of human readers reading the text and use them as features along with textual features for the tasks of sentiment polarity and sarcasm detection. Our proposed framework is based on Convolutional Neural Network (CNN). The CNN *learns features* from both gaze and text and uses them to classify the input text. We test our technique on published sentiment and sarcasm labeled datasets, enriched with gaze information, to show that using a combination of automatically learned text and gaze features yields better classification performance over (i) CNN based systems that rely on text input alone and (ii) existing systems that rely on handcrafted gaze and textual features.

## 1 Introduction

Detection of sentiment and sarcasm in user-generated short reviews is of primary importance for social media analysis, recommendation and dialog systems. Traditional sentiment analyzers and sarcasm detectors face challenges that arise at *lexical*, *syntactic*, *semantic* and *pragmatic* levels (Liu and Zhang, 2012; Mishra et al., 2016c). Feature-based systems (Akkaya et al., 2009; Sharma and Bhattacharyya, 2013; Poria et al., 2014) can aptly handle lexical and syntactic challenges (*e.g.* learning that the word *deadly* conveys a strong positive sentiment in opinions such as *Shane Warne is a deadly bowler*, as opposed to *The high altitude Himalayan roads have deadly turns*). It is, however, extremely difficult to tackle subtleties at semantic and pragmatic levels. For example, the sentence *"I really love my job. I work 40 hours a week to be this poor."* requires an NLP system to be able to understand that the opinion holder has not expressed a positive sentiment towards her / his job. In the absence of explicit clues in the text, it is difficult for automatic systems to arrive at a correct classification decision, as they often lack external knowledge about various aspects of the text being classified.

Mishra et al. (2016b) and Mishra et al. (2016c) show that NLP systems based on cognitive data (or simply, *Cognitive NLP* systems) , that of leverage eye-movement information obtained from human readers, can tackle the semantic and pragmatic challenges better. The hypothesis here is that human gaze activities are related to the cognitive processes in the brain, that combines the "external knowledge" that the a reader possesses with textual clues that she / he perceives. While incorporating behavioral information obtained from gaze-data in NLP systems is intriguing and quite plausible, especially due to the availability of low cost eye-tracking machinery (Wood and Bulling, 2014; Yamamoto et al., 2013), few methods exist for text classification and they rely on handcrafted features extracted from gaze data (Mishra et al., 2016b,c). These systems have limited capabilities due to two reasons: (a) Manually designed gaze based features may not adequately capture all

forms of textual subtleties (b) Eye-movement data is not as intuitive to analyze as text which makes the task of designing manual features more difficult. So, in this work, **instead of handcrafting the gaze based and textual features, we try to learn feature representations from both gaze and textual data using Convolutional Neural Networks (CNNs)**. We test our technique on two publicly available datasets enriched with eye-movement information, used for *binary classification* tasks of sentiment polarity and sarcasm detection. Our experiments show that the automatically extracted features help to achieve significant classification performance improvement over (a) existing systems that rely on handcrafted gaze and textual features and (b) CNN based systems that rely on text input alone.

The rest of the paper is organized as follows. Section 2 discusses the motivation behind using readers' eye-movement data in a text classification setting. Section 3 discusses on why CNNs is preferred over other available alternatives for feature extraction. The CNN architecture is proposed and discussed in Section 4. Section 5 describes our experimental setup and results are discussed in Section 6. We provide a detailed analysis of the results along with some insightful observations in Section 7. Section 8 points to relevant literature followed by Section 9 that concludes the paper.

**Terminology:** A *fixation* is a relatively long stay of gaze on a visual object (such as words in text) where as a *sacccade* corresponds to quick shifting of gaze between two positions of rest. Forward and backward saccades are called *progressions* and *regressions* respectively. A *scanpath* is a line graph that contains fixations as nodes and saccades as edges.

## 2 Eye-movement and Linguistic Subtleties

Presence of linguistic subtleties often induces (a) surprisal (Kutas and Hillyard, 1980; Malsburg et al., 2015), due to the underlying disparity /context incongruity or (b) higher cognitive load (Rayner and Duffy, 1986), due to the presence of lexically and syntactically complex structures. While surprisal accounts for irregular saccades (Malsburg et al., 2015), higher cognitive load results in longer fixation duration (Kliegl et al., 2004).

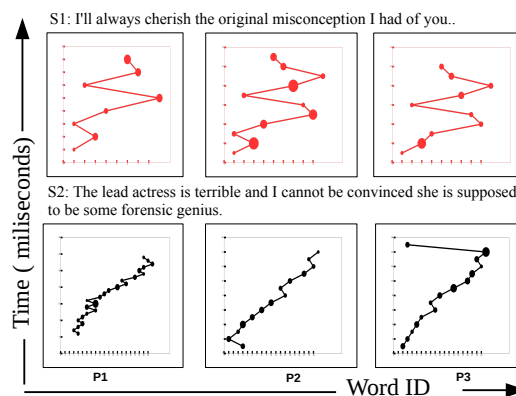

Figure 1: Scanpaths of three participants for two sentences (Mishra et al., 2016b). Sentence *S1* is sarcastic but *S2* is not. Length of the straight lines represents saccade distance and size of the circles represents fixation duration

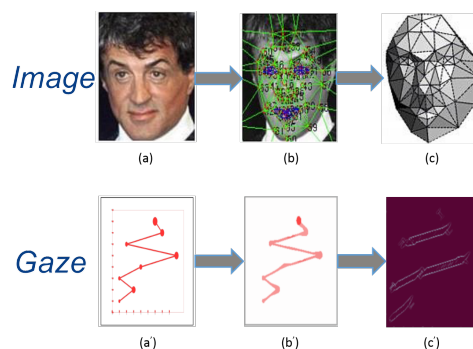

Figure 2: Illustrative analogy between CNN applied to images and scanpath representation showing why CNN can be useful for learning features from gaze patterns

Mishra et al. (2016b) find that presence of sarcasm in text triggers either *irregular saccadic patterns* or *unusually high duration fixations* than non-sarcastic texts (illustrated through example scanpath representations in Figure 1). For sentiment bearing texts, highly subtle eye-movement patterns are observed for semantically / pragmatically complex negative opinions (expressing irony, sarcasm, thwarted expectations etc.) than the simple ones (Mishra et al., 2016b). The association between linguistic subtleties and eye-movement patterns could be captured through sophisticated feature engineering that considers both gaze and text inputs. In our work, CNNs take the onus of feature engineering.

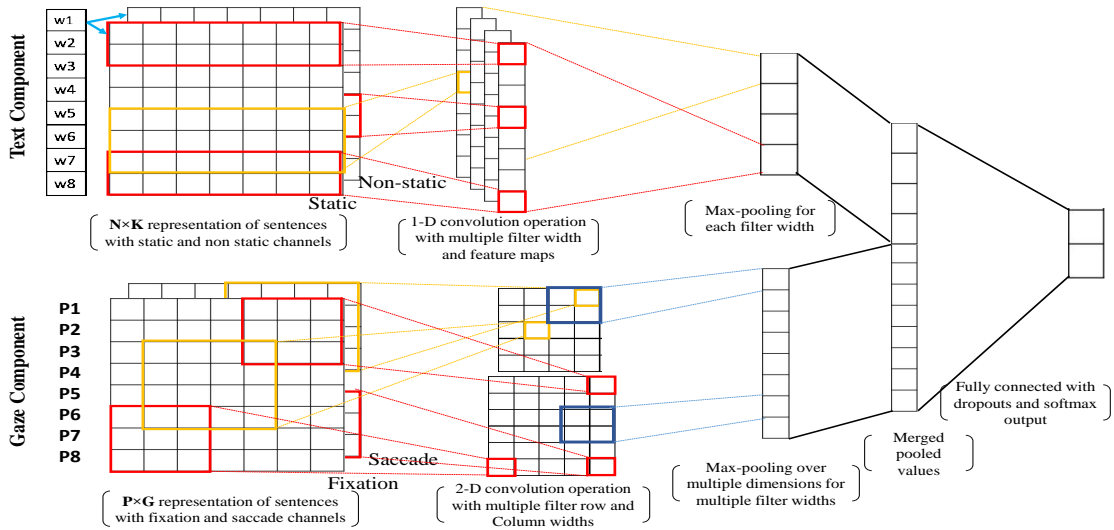

Figure 3: Deep convolutional model for feature extraction from both text and gaze inputs

## 3 Why Convolutional Neural Network?

CNNs have been quite effective in learning *filters* for image processing tasks, filters being used to transform the input image into more informative feature space (Krizhevsky et al., 2012). Filters learned at various CNN layers are quite similar to handcrafted filters used for detection of edges, contours and removal of redundant backgrounds. We believe, a similar technique can also be applied to eye-movement data, where the learned filters will, hopefully, extract informative cognitive features. For instance, for sarcasm, we expect the network to learn filters that detect long distance saccades (refer to Figure 2 for an analogical illustration). With more number of convolution filters of different dimensions, the network may extract multiple features related to different gaze attributes (such as fixations, progressions, regressions and skips) and will be free from any form of human bias that manually extracted features are susceptible to.

## 4 Learning Feature Representations: The CNN architecture

Figure 3 shows the CNN architecture with two components for processing and extracting features from text and gaze inputs. The components are explained below.

### 4.1 Text Component

The text component is quite similar to the one proposed by Kim (2014) for sentence classification.

Words (in the form of *one-hot* representation) in the input text are first replaced by their embeddings of dimension $K$ ($i^{th}$ word in the sentence represented by an embedding vector $x_i \in \mathbb{R}^K$). As per Kim (2014), a multi-channel variant of CNN (referred to as MULTICHANNELTEXT) can be implemented by using two channels of embeddings-one that remains static through out training (referred to as STATICTEXT), and the other one that gets updated during training (referred to as NON-STATICTEXT). We separately experiment with static, non-static and multi-channel variants.

For each possible input channel of the text component, a given text is transformed into a tensor of fixed length $N$ (padded with *zero-tensors* wherever necessary to tackle length variations) by concatenating the word embeddings.

$$x_{1:N} = x_1 \oplus x_2 \oplus x_3 \oplus ... \oplus x_N \quad (1)$$

where $\oplus$ is the concatenation operator. To extract *local features*[1], convolution operation is applied. Convolution operation involves a *filter*, $W \in \mathbb{R}^{HK}$, which is convolved with a window of $H$ embeddings to produce a local feature for the $H$ words. A local feature, $c_i$ is generated from a window of embeddings $x_{i:i+H-1}$ by applying a non linear function (such as a hyperbolic tangent) over the convoluted output. Mathematically,

$$c_i = f(W.x_{i:i+H-1} + b) \quad (2)$$

---
[1]features specific to a region in case of images or window of words in case of text

where $b \in \mathbb{R}$ is the *bias* and $f$ is the non-linear function. This operation is applied to each possible window of $H$ words to produce a feature map ($c$) for the window size $H$.

$$c = [c_1, c_2, c_3, ..., c_{N-H+1}] \qquad (3)$$

A global feature is then obtained by applying *max pooling* operation[2] (Collobert et al., 2011) over the feature map. The idea behind *max-pooling* is to capture the most important feature - one with the highest value - for each feature map.

We have described the process by which one feature is extracted from one filter (for illustration, red bordered portions in Figure 3 depict the case of $H = 2$). The model uses multiple filters (with varying window sizes) to obtain multiple features representing the text. In the MULTICHANNEL-TEXT variant, for a window of $H$ words, convolution operation is separately applied on both the embedding channels. Local features learned from both the channels are concatenated before applying *max-pooling*.

## 4.2 Gaze Component

The gaze component deals with scanpaths of multiple participants annotating the same text. Scanpaths can be pre-processed to extract two sequences of gaze data to form separate channels of input: (1) A sequence of normalized[3] durations of fixations (in milliseconds) in the order in which they appear in the scanpath and (2) A sequence of position of fixations (in terms of word id) in the order in which they appear in the scanpath. These channels are related to two fundamental gaze attributes such as fixation and saccade respectively. With two channels, we thus have three possible configurations of the gaze component such as (i) FIXATION, where the input is normalized fixation duration sequence, (ii) SACCADE, where the input is fixation position sequence, and (iii) MULTI-CHANNELGAZE, where both the inputs channels are considered.

For each possible input channel, the input is in the form of a $P \times G$ matrix (with $P \to$ number of participants and $G \to$ length of the input sequence). Each element of the matrix $g_{ij} \in \mathbb{R}$, with $i \in P$ and $j \in G$, corresponds to the $j^{th}$ gaze attribute (either fixation duration or word id, depending on the channel) of the input sequence of

the $i^{th}$ participant. Now, unlike the text component, here we apply convolution operation across two dimensions *i.e.* choosing a two dimensional convolution filter $W \in \mathbb{R}^{JK}$ (for simplicity, we have kept $J = K$, thus, making the dimension of $W$, $J^2$). For the dimension size of $J^2$, a local feature $c_{ij}$ is computed from the window of gaze elements $g_{ij:(i+J-1)(j+J-1)}$ by,

$$c_{ij} = f(W.g_{ij:(i+J-1)(j+J-1)} + b) \qquad (4)$$

where $b \in \mathbb{R}$ is the *bias* and $f$ is a non-linear function. This operation is applied to each possible window of size $J^2$ to produce a feature map ($c$),

$$\begin{aligned} c = [&c_{11}, c_{12}, c_{13}, ..., c_{1(G-J+1)}, \\ &c_{21}, c_{22}, c_{23}, ..., c_{2(G-J+1)}, \\ &..., \\ &c_{(P-J+1)1}, c_{(P-J+1)2}, ..., c_{(P-J+1)(G-J+1)}] \end{aligned} \qquad (5)$$

A global feature is then obtained by applying *max pooling* operation. Unlike the text component, max-pooling operator is applied to a 2D window of local features size $M \times N$ (for simplicity, we set $M = N$, denoted henceforth as $M^2$). For the window of size $M^2$, the pooling operation on $c$ will result in as set of global features $\hat{c}_J = max\{c_{ij:(i+M-1)(j+M-1)}\}$ for each possible $i, j$.

We have described the process by which one feature is extracted from one filter (of 2D window size $J^2$ and max-pooling window size of $M^2$). In Figure 3, red and blue bordered portions illustrate the cases of $J^2 = [3, 3]$ and $M^2 = [2, 2]$ respectively. Like the text component, the gaze component uses multiple filters (also with varying window size) to obtain multiple features representing the gaze input. In the MULTICHANNEL-GAZE variant, for a 2D window of $J^2$, convolution operation is separately applied on both fixation duration and saccade channels and local features learned from both the channels are concatenated before max-pooling is applied.

Once the global features are learned from both the text and gaze components, they are *merged* and passed to a fully connected feed forward layer (with number of units set to 150) followed by a *SoftMax* layer that outputs the the probabilistic distribution over the class labels.

The gaze component of our network is not invariant of the order in which the scanpath data is given as input- *i.e.*, the $P$ rows in the $P \times G$ can not be shuffled, even if each row is independent from others. The only way we can think of for

---

[2]*mean pooling* does not perform well.

[3]scaled across participants using min-max normalization to reduce subjectivity

addressing this issue is by applying convolution operations to all $P \times G$ matrices formed with all the permutations of $P$, capturing every possible ordering. Unfortunately, this makes the training process significantly less scalable, as the number of model parameters to be learned becomes huge. As of now, training and testing are carried out by keeping the order of the input constant.

## 5 Experiment Setup

We now share several details regarding our experiments below.

**1. Dataset:** We experiment on sentiment and sarcasm tasks using two publicly available datasets enriched with eye-movement information. Dataset 1 has been released by Mishra et al. (2016a). It contains 994 text snippets with 383 positive and 611 negative examples. Out of the 994 snippets, 350 are sarcastic. Dataset 2 has been used by Joshi et al. (2014) and it consists of 843 snippets comprising movie reviews and normalized tweets out of which 443 are positive and 400 are negative. Eye-movement data of 7 and 5 readers is available for each snippet for dataset 1 and 2 respectively.

**2. CNN Variants:** With text component alone we have three variants such as STATICTEXT, NON-STATICTEXT and MULTICHANNELTEXT (refer to Section 4.1). Similarly, with gaze component we have variants such as FIXATION, SACCADE and MULTICHANNELGAZE (refer to Section 4.2). With both text and gaze components, 9 more variants could be experimented with.

**3. Hyper-parameters:** For text component, we experiment with filter widths ($H$) of $[3, 4]$. For the gaze component, 2D filters ($J^2$) set to $[3 \times 3], [4 \times 4]$ respectively. The max pooling 2D window, $M^2$, is set to $[2 \times 2]$. In both gaze and text components, number of filters is set to 150, resulting in 150 feature maps for each window. These model hyper-parameters are fixed by trial and error and are possibly good enough to provide a first level insight into our system. Tuning of hyper-parameters might help in improving the performance of our framework, which is on our future research agenda.

**4. Regularization:** For regularization *dropout* is employed on the penultimate layer with a constraint on $l_2$-norms of the weight vectors (Hinton et al., 2012). Dropout prevents co-adaptation of hidden units by randomly dropping out - i.e., setting to zero - a proportion $p$ of the hidden units

during forward propagation. We set $p$ to 0.25.

**5. Training:** We use ADADELTA optimizer (Zeiler, 2012), with a learning rate of 0.1. The input batch size is set to 32 and number of training iterations (epochs) is set to 200. 10% of the training data is used for validation.

**6. Use of pre-trained embeddings:** Initializing the embedding layer with of pre-trained embeddings can be more effective than random initialization (Kim, 2014). In our experiments, we have used embeddings using *word2vec* facilitated by Mikolov et al. (2013) (best results obtained with embedding dimension of 50). We have also tried randomly initializing the embeddings but better results are obtained with pre-trained embeddings.

**7. Comparison with existing work:** For sentiment analysis, we compare our systems's accuracy (for both datasets 1 and 2) with Mishra et al. (2016c)'s systems that rely on handcrafted text and gaze features. For sarcasm detection, we compare Mishra et al. (2016b)'s sarcasm classifier with ours using dataset 1 (with available gold standard labels for sarcasm). We follow the same 10-fold train-test configuration as these existing works for consistency.

## 6 Results

In this section, we discuss the results for different model variants for sentiment polarity and sarcasm detection tasks.

### 6.1 Results for Sentiment Analysis Task

Table 1 presents results for sentiment analysis task. For dataset 1, different variants of our CNN architecture outperform the best systems reported by Mishra et al. (2016c), with a maximum F-score improvement of 3.8%. This improvement is statistically significant of $p < 0.05$ as confirmed by **McNemar test**. Moreover, we observe an F-score improvement of around 5% for CNNs with both gaze and text components as compared to CNNs with only text components (similar to the system by Kim (2014)), which is also statistically significant (with $p < 0.05$).

For dataset 2, CNN based approaches do not perform better than manual feature based approaches. However, variants with both text and gaze components outperform the ones with only text component (Kim, 2014), with a maximum F-score improvement of 2.9%. We observe that for dataset 2, training accuracy reaches 100 within

| | Configuration | Dataset1 | | | Dataset2 | | |
|---|---|---|---|---|---|---|---|
| | | P | R | F | P | R | F |
| Traditional systems based on textual features | Näive Bayes | 63.0 | 59.4 | 61.14 | 50.7 | 50.1 | 50.39 |
| | Multi-layered Perceptron | 69.0 | 69.2 | 69.2 | 66.8 | 66.8 | 66.8 |
| | SVM (Linear Kernel) | 72.8 | 73.2 | 72.6 | 70.3 | 70.3 | 70.3 |
| Systems by Mishra et al. (2016c) | Gaze based (Best) | 61.8 | 58.4 | 60.05 | 53.6 | 54.0 | 53.3 |
| | Text + Gaze (Best) | **73.3** | **73.6** | **73.5** | **71.9** | **71.8** | **71.8** |
| CNN with only text input (Kim, 2014) | STATICTEXT | 63.85 | 61.26 | 62.22 | 55.46 | 55.02 | 55.24 |
| | NONSTATICTEXT | 72.78 | 71.93 | 72.35 | 60.51 | 59.79 | 60.14 |
| | MULTICHANNELTEXT | 72.17 | 70.91 | 71.53 | 60.51 | 59.66 | 60.08 |
| CNN with only gaze Input | FIXATION | 60.79 | 58.34 | 59.54 | 53.95 | 50.29 | 52.06 |
| | SACCADE | 64.19 | 60.56 | 62.32 | 51.6 | 50.65 | 51.12 |
| | MULTICHANNELGAZE | 65.2 | 60.35 | 62.68 | 52.52 | 51.49 | 52 |
| CNN with both text and gaze Input | STATICTEXT + FIXATION | 61.52 | 60.86 | 61.19 | 54.61 | 54.32 | 54.46 |
| | STATICTEXT + SACCADE | 65.99 | 63.49 | 64.71 | 58.39 | 56.09 | 57.21 |
| | STATICTEXT + MULTICHANNELGAZE | 65.79 | 62.89 | 64.31 | 58.19 | 55.39 | 56.75 |
| | NONSTATICTEXT + FIXATION | 73.01 | 70.81 | 71.9 | 61.45 | 59.78 | 60.60 |
| | NONSTATICTEXT + SACCADE | 77.56 | 73.34 | 75.4 | **65.13** | **61.08** | **63.04** |
| | NONSTATICTEXT + MULTICHANNELGAZE | **79.89** | **74.86** | **77.3** | 63.93 | 60.13 | 62 |
| | MULTICHANNELTEXT + FIXATION | 74.44 | 72.31 | 73.36 | 60.72 | 58.47 | 59.57 |
| | MULTICHANNELTEXT + SACCADE | **78.75** | **73.94** | **76.26** | 63.7 | 60.47 | 62.04 |
| | MULTICHANNELTEXT + MULTICHANNELGAZE | **78.38** | **74.23** | **76.24** | 64.29 | 61.08 | 62.64 |

Table 1: Results for different traditional feature based systems and CNN model variants for the task of sentiment analysis. Abbreviations (P,R,F)→ Precision, Recall, F-score. SVM→Support Vector Machine

25 epochs with validation accuracy stable around $50\%$, indicating the possibility of overfitting. Tuning the regularization parameters specific to dataset 2 may help here. Even though CNN might not be proving to be a choice as good as hand-crafted features for dataset 2, the bottom line remains that incorporation of gaze data into CNN consistently improves the performance over only-text-based CNN variants.

## 6.2 Results for Sarcasm Detection Task

For sarcasm detection, our CNN model variants outperform traditional systems by a maximum margin of $11.27\%$ (Table 2). However, the improvement by adding the gaze component to the CNN network is just $1.36\%$, which is statistically insignificant over CNN with text component. While inspecting the sarcasm dataset, we observe a clear difference between the vocabulary of sarcasm and non-sarcasm classes in our dataset. This, perhaps, was captured well by the text component, especially the variant with only non-static embeddings.

## 7 Discussion

In this section, some important observations from our experiments are discussed.

• **Effect of embedding dimension variation:** Embedding dimension has proven to have a deep impact on the performance of neural systems (dos Santos and Gatti, 2014; Collobert et al., 2011). We repeated our experiments by varying the embedding dimensions in the range of [50-300][4] and observed that reducing embedding dimension improves the F-scores by a little margin. Best results are obtained when the embedding dimension is as low as $50$. Small embedding dimensions are probably reducing the chances of over-fitting when the data size is small. We also observe that for different embedding dimensions, performance of CNN with both gaze and text components is consistently better than that with only text component.

• **Effect of static / non static text channels:** Non-static embedding channel has a major role in tuning embeddings for sentiment analysis by bringing adjectives expressing similar sentiment close to each other (*e.g, good and nice*), where as static channel seems to prevent over-tuning of embeddings (over-tuning often brings verbs like *love* closer to the pronoun *I* in embedding space, purely due to higher co-occurrence of these two words in sarcastic examples).

• **Effect of fixation / saccade channels:** For sentiment detection, saccade channel seems to be handing text having semantic incongruity (due to the presence of irony / sarcasm) better. Fixation channel does not help much, may be because of higher variance in fixation duration. For sarcasm

---

[4]a standard range (Liu et al., 2015; Melamud et al., 2016)

| | Configuration | P | R | F |
|---|---|---|---|---|
| Traditional systems based on textual features | Näive Bayes | 69.1 | 60.1 | 60.5 |
| | Multi-layered Perceptron | 69.7 | 70.4 | 69.9 |
| | SVM (Linear Kernel) | 72.1 | 71.9 | 72 |
| Systems by Riloff et al. (2013) | Text based (Ordered) | 49 | 46 | 47 |
| | Text + Gaze (Unordered) | 46 | 41 | 42 |
| System by Joshi et al. (2015) | Text based (best) | 70.7 | 69.8 | 64.2 |
| Systems by Mishra et al. (2016b) | Gaze based (Best) | 73 | 73.8 | 73.1 |
| | Text based (Best) | 72.1 | 71.9 | 72 |
| | Text + Gaze (Best) | 76.5 | 75.3 | 75.7 |
| CNN with only text input (Kim, 2014) | STATICTEXT | 67.17 | 66.38 | 66.77 |
| | NONSTATICTEXT | 84.19 | **87.03** | 85.59 |
| | MULTICHANNELTEXT | 84.28 | **87.03** | 85.63 |
| CNN with only gaze input | FIXATION | 74.39 | 69.62 | 71.93 |
| | SACCADE | 68.58 | 68.23 | 68.40 |
| | MULTICHANNELGAZE | 67.93 | 67.72 | 67.82 |
| CNN with both text and gaze Input | STATICTEXT + FIXATION | 72.38 | 71.93 | 72.15 |
| | STATICTEXT + SACCADE | 73.12 | 72.14 | 72.63 |
| | STATICTEXT + MULTICHANNELGAZE | 71.41 | 71.03 | 71.22 |
| | NONSTATICTEXT + FIXATION | **87.42** | 85.2 | 86.30 |
| | NONSTATICTEXT + SACCADE | 84.84 | 82.68 | 83.75 |
| | NONSTATICTEXT + MULTICHANNELGAZE | 84.98 | 82.79 | 83.87 |
| | MULTICHANNELTEXT + FIXATION | 87.03 | 86.92 | **86.97** |
| | MULTICHANNELTEXT + SACCADE | 81.98 | 81.08 | 81.53 |
| | MULTICHANNELTEXT + MULTICHANNELGAZE | 83.11 | 81.69 | 82.39 |

Table 2: Results for different traditional feature based systems and CNN model variants for the task of sarcasm detection on dataset 1. Abbreviations (P,R,F)→ Precision, Recall, F-score

detection, fixation and saccade channels perform with similar accuracy when employed separately. Accuracy reduces with gaze multichannel, may be because of higher variation of both fixations and saccades across sarcastic and non-sarcastic classes, as opposed to sentiment classes.

• **Effectiveness of the CNN learned features** To examine how good the features learned by the CNN are, we analyzed the features for a few example cases. Figure 4 presents some of the test-examples for the task of sarcasm detection. Example 1 contains sarcasm while examples 2, 3 and 4 are non-sarcastic. To see if there is any difference in the automatically learned features between text-only and combined text and gaze variants, we examine the feature vector (of dimension 150) for the examples obtained from different model variants. Output of the hidden layer after *merge* layer is considered as features learned by the network. We plot the features, in the form of color-bars, following Li et al. (2016) - denser colors representing higher feature values. In Figure 4, we show only two (representative) model variants *viz.*, MULTICHANNELTEXT and MULTICHANNELTEXT+ MULTICHANNELGAZE. As one can see, addition of gaze information helps

to generate features with more subtle differences (marked by blue rectangular boxes) for sarcastic and non-sarcastic texts. It is also interesting to note that in the marked region, features for the sarcastic texts exhibit more intensity than the non-sarcastic ones - perhaps capturing the notion that sarcasm typically conveys an intensified negative opinion. This difference is not clear in feature vectors learned by text only systems for instances like example 2, which has been incorrectly classified by MULTICHANNELTEXT. Example 4 is incorrectly classified by both the systems, perhaps due to lack of context. In cases like this, addition of gaze information does not help much in learning more distinctive features, as it becomes difficult for even humans to classify such texts.

## 8 Related Work

Sentiment and sarcasm classification are two important problems in NLP and have been the focus of research for many communities for quite some time. Popular sentiment and sarcasm detection systems are feature based and are based on unigrams, bigrams etc. (Dave et al., 2003; Ng et al., 2006), syntactic properties (Martineau and Finin, 2009; Nakagawa et al., 2010), semantic properties

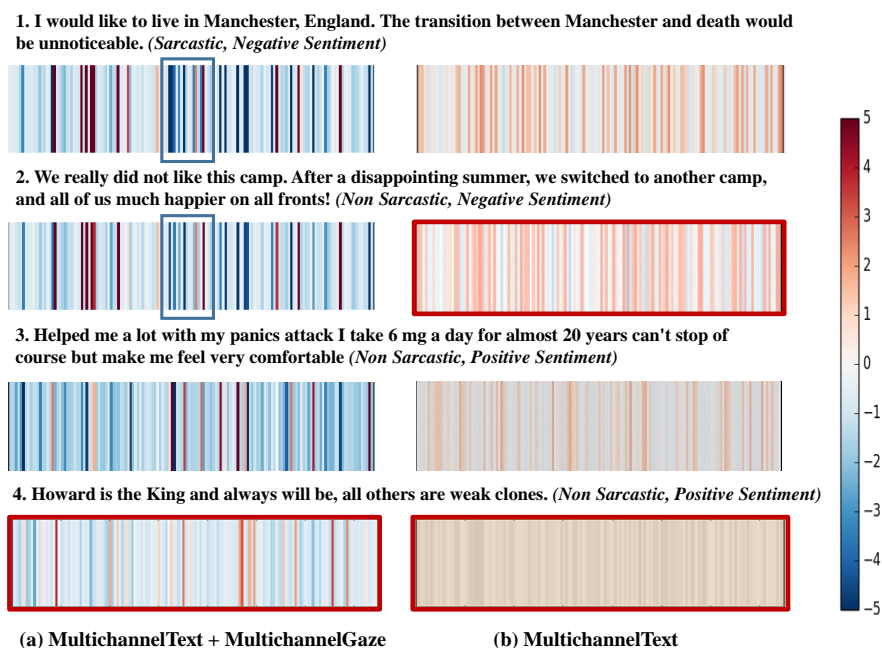

1. I would like to live in Manchester, England. The transition between Manchester and death would be unnoticeable. *(Sarcastic, Negative Sentiment)*

2. We really did not like this camp. After a disappointing summer, we switched to another camp, and all of us much happier on all fronts! *(Non Sarcastic, Negative Sentiment)*

3. Helped me a lot with my panics attack I take 6 mg a day for almost 20 years can't stop of course but make me feel very comfortable *(Non Sarcastic, Positive Sentiment)*

4. Howard is the King and always will be, all others are weak clones. *(Non Sarcastic, Positive Sentiment)*

**(a) MultichannelText + MultichannelGaze**  **(b) MultichannelText**

Figure 4: Visualization of representations learned by two variants of the network for sarcasm detection task. The output of the *Merge* layer (of dimension 150) are plotted in the form of colour-bars. Plots with thick red borders correspond to wrongly predicted examples.

(Balamurali et al., 2011). For sarcasm detection, supervised approaches rely on **(a)** Unigrams and Pragmatic features (González-Ibánez et al., 2011; Barbieri et al., 2014; Joshi et al., 2015) **(b)** Stylistic patterns (Davidov et al., 2010) and patterns related to *situational disparity* (Riloff et al., 2013) and **(c)** Hastag interpretations (Liebrecht et al., 2013; Maynard and Greenwood, 2014). Recent systems are based on variants of deep neural network built on the top of embeddings. A few representative works in this direction for sentiment analysis are based on CNNs (dos Santos and Gatti, 2014; Kim, 2014; Tang et al., 2014), RNNs (Dong et al., 2014; Liu et al., 2015) and combined architecture (Wang et al., 2016). Few works exist on using deep neural networks for sarcasm detection, one of which is by (Ghosh and Veale, 2016) that uses a combination of RNNs and CNNs.

Eye-tracking technology is a relatively new NLP, with a very few systems directly making use of gaze data in prediction frameworks. Klerke et al. (2016) present a novel multi-task learning approach for sentence compression using labeled data, while, Barrett and Søgaard (2015) discriminate between grammatical functions using gaze features. The closest works to ours is by Mishra et al. (2016b) and Mishra et al. (2016c) that introduce feature engineering based on both gaze

and text data for sentiment and sarcasm detection tasks. These recent advancements motivate us to explore the cognitive NLP paradigm .

# 9 Conclusion and Future Directions

In this work, we proposed a multimodal ensemble of features, automatically learned using variants of CNNs from text and readers' eye-movement data, for the tasks of sentiment and sarcasm classification. On multiple published datasets for which gaze information is available, our systems could achieve significant performance improvements over (a) systems that rely on handcrafted gaze and textual features and (b) CNN based systems that rely on text input alone. An analysis of the learned features confirms that the combination of automatically learned features is indeed capable of representing deep linguistic subtleties in text that pose challenges to sentiment and sarcasm classifiers. Our future agenda includes: (a) optimizing the CNN framework hyper-parameters (e.g., filter width, dropout, embedding dimensions etc.) to obtain better results, (b) exploring the applicability of our technique for document-level sentiment analysis and (c) applying our framework on related problems, such as emotion analysis, text summarization and question-answering.

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
