# Peer review of "Learning Cognitive Features from Gaze Data for Sentiment and Sarcasm Classification using Convolutional Neural Network"

_ACL 2017 — decision unknown_

[Official Review · Reviewer 1 · rating 4 · confidence 4]
soundness 5 · originality 5 · clarity 5 · impact 3 · substance 4 · appropriateness 5 · meaningful comparison 3 · presentation format Oral Presentation

- Strengths:

This paper tackles an interesting problem and provides a (to my knowledge)
novel and reasonable way of learning and combining cognitive features with
textual features for sentiment analysis and irony detection. The paper is 
clearly written and organized, and the authors provided a lot of useful detail
and informative example and plots. Most of the results are convincing, and the
authors did a good job comparing their approach and results with previous work.

- Weaknesses:

1. Just from the reading abstract, I expected that the authors' approach would
significantly outperform previous methods, and that using both the eye-gaze and
textual features consistently yields the best results. Upon reading the actual
results section, however, it seems like the findings were more mixed. I think
it would be helpful to update the abstract and introduction to reflect this. 
2. When evaluating the model on dataset 1 for sentiment analysis, were the
sarcastic utterances included? Did the model do better on classifying the
non-sarcastic utterances than the sarcastic ones?
3. I understand why the eye-movement data would be useful for sarcasm
detection, but it wasn't as obvious to me why it would be helpful for
(non-sarcastic) sentiment classification beyond the textual features. 

- General Discussion:

This paper contains a lot of interesting content, and the approach seems solid
and novel to me. The results were a little weaker than I had anticipated from
the abstract, but I believe would still be interesting to the larger community
and merits publication.

[Official Review · Reviewer 2 · rating 3 · confidence 3]
soundness 5 · originality 5 · clarity 3 · impact 3 · substance 4 · appropriateness 5 · meaningful comparison 3 · presentation format Oral Presentation

- Strengths:

(1) A deep CNN framework is proposed to extract and combine cognitive features
with textual features for sentiment analysis and sarcasm detection. 

(2) The ideas is interesting and novelty.

- Weaknesses:

(1) Replicability would be an important concern. Researchers cannot replicate
the system/method for improvement due to lack of data for feature extraction. 

- General Discussion:

Overall, this paper is well written and organized. The experiments are
conducted carefully for comparison with previous work and the analysis is
reasonable. I offer some comments as follows.

(1)           Does this model be suitable on sarcastic/non-sarcastic utterances?
The
authors should provide more details for further analysis. 

(2)           Why the eye-movement data would be useful for
sarcastic/non-sarcastic
sentiment classification beyond the textual features? The authors should
provide more explanations.